# Novelty Detection in Reinforcement Learning with World Models

**Geigh Zollicoffer** [1]  **Kenneth Eaton** [2]  **Jonathan Balloch** [3]  **Julia Kim** [3]  **Wei Zhou** [3]  **Robert Wright** [2]  **Mark Riedl** [3]

## Abstract

Reinforcement learning (RL) using world models has found significant recent successes. However, when a sudden change to world mechanics or properties occurs then agent performance and reliability can dramatically decline. We refer to the sudden change in visual properties or state transitions as *novelties*. Implementing novelty detection within generated world model frameworks is a crucial task for protecting the agent when deployed. In this paper, we propose straightforward bounding approaches to incorporate novelty detection into world model RL agents by utilizing the misalignment of the world model's hallucinated states and the true observed states as a novelty score. We provide effective approaches to detecting novelties in a distribution of transitions learned by an agent in a world model. Finally, we show the advantage of our work in Mini-Grid, Atari, and DeepMind Control environments compared to traditional machine learning novelty detection methods as well as currently accepted RL-focused novelty detection algorithms.

## 1. Introduction

Reinforcement learning (RL) using world models has found significant recent successes due to its strength in sampling efficiency and ability to incorporate well-studied Markov Decision Process techniques (Moerland et al., 2020; Robine et al., 2021). A *world model* (Ha & Schmidhuber, 2018b) is a model that predicts the world state given a current state and the execution that was executed (or will be executed).

While RL agents are often trained and evaluated in environments with stationary transition functions, the real world can undergo distributional shifts in the underlying transition dynamics. In this paper, we address novelty detection, a form of anomaly detection, which is relatively unexplored in reinforcement learning (Müller et al., 2022; Nasvytis et al., 2024). *Novelties* are sudden changes to the observation space or environment state transition dynamics that occur *at inference time* that are unanticipated (or unanticipatable) by the agent during training (Balloch et al., 2022). Novelties represent a *permanent* shift in observation space or environment state transition dynamics, as opposed to one-off sensory or transition errors; when a novelty occurs it may not be encountered immediately by an agent, and because transition dynamics may be uncertain, only become apparent after several actions (Balloch et al., 2022).

To motivate the work, consider Table 1. The top row shows what an agent sees when training, when a novelty is introduced at inference time, and post-novelty. The bottom row shows the reconstructed image from the world model's prediction of the next state. Post-novelty, the world model's predicted state begins differing radically from the ground truth. In cases where there is a novel change, the RL agent's converged policy is no longer reliable, and the agent can make catastrophic mistakes. The agent may flounder ineffectually, or, worse, the agent can mistakenly take action that put itself or others in harms way.

There are several ways of addressing inference-time novelty depending on the nature of the agent's task. One may halt execution because continuing to run the policy has become potentially dangerous, for example in the case of a robotic platform or agent interacting with people, financial systems, or other high-stakes situations. Once halted, an operator figures out the best way to retrain the agent. Alternatively, one may attempt *novelty adaptation* where the agent attempts to update its own policy during online inference time (Zhao et al., 2019; Wilson & Cook, 2020).

In either case, one must *detect* that novelty has occurred prior to halting or adapting. The standard approach—as exemplified by Recurrent Implicit Quantile Networks (RIQN)[1] (Danesh & Fern, 2021)—is to treat novelty as a distribution shift in either the observation space or the state-action transition probability. This is traditionally accom-

---

[1]Department of Mathematics, Georgia Institute of Technology, Atlanta, United States of America [2]Georgia Tech Research Institute, Atlanta, United States of America [3]Department of Computer Science, Georgia Institute of Technology, Atlanta, United States of America. Correspondence to: Mark Riedl <riedl@cc.gatech.edu>, Geigh Zollicoffer <gzollicoffer3@gatech.edu>.

*Proceedings of the $42^{nd}$ International Conference on Machine Learning*, Vancouver, Canada. PMLR 267, 2025. Copyright 2025 by the author(s).

---

[1]RIQN is the baseline method used in Out-of-Distribution Dynamics Detection: RL-Relevant Benchmarks and Results.

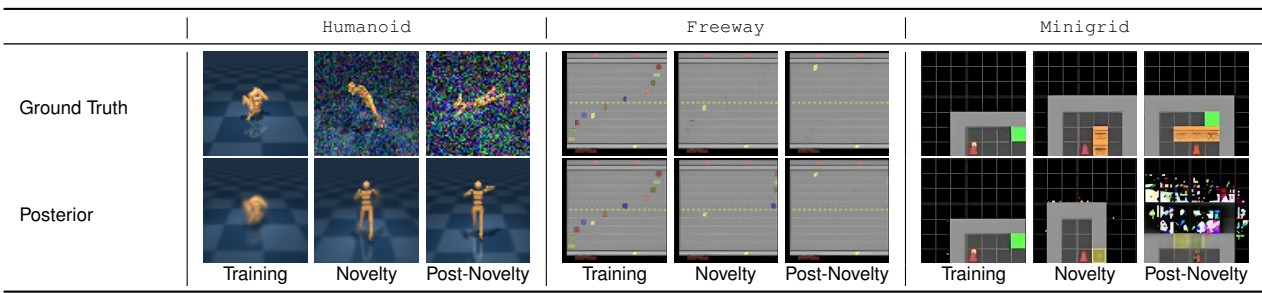

Table 1: Examples of model collapse when encountering novelty in three environments: DeepMind Control Suite Humanoid (noise introduced), Atari Freeway (cars become invisible), and MiniGrid (lava introduced). The *Ground Truth* row shows actual screens during training, when a novelty is introduced, and post-novelty. The *posterior* row shows the world model's reconstructed prediction, which deviates from the ground truth.

plished by setting a threshold hyperparameter to determine when the distribution shift is significant enough to be considered a novelty. Setting such a parameter requires at least implicit foreknowledge of anomalies that can occur, which violates our assumptions that novelties be unanticipatable and unencountered during training.

We introduce a novel technique for novelty detection *without* the need to manually specify thresholds, and *without* the need for additional augmented data. Our approach builds on the capabilities of world models to predict the state of the world given the current state and an action. When a novelty occurs, our insight is that the predicted state and actual state after an action will deviate. Our technique calculates a **novelty threshold bound** without additional hyper-parameters by considering how much the actual world observation deviates from the distribution of world observations that the agent predicts it will encounter.

Our technique draws from the notion of *Bayesian surprise* (Itti & Baldi, 2005). Specifically, as the agent interacts with the environment, it tracks the KL divergence between the predicted latent world state representation given a hidden state and embedded image relative to the predicted latent world state given just the hidden state (without the embedded image). To develop the bound, we observe that, under nominal conditions, any divergence should be smaller than that of the predicted latent world state computed with the initial hidden state, as the latter prediction becomes increasingly inaccurate. But, as we will show, novelty can flip this relationship. When this happens, we flag the violation of an inequality of divergences.

We evaluate our method by injecting novelties into Mini-Grid (Chevalier-Boisvert et al., 2018), Atari (Machado et al., 2018), and continuous DeepMind Control (DMC) (Tunyasu-vunakool et al., 2020) environments. Specifically, we use the NovGrid (Balloch et al., 2022), HackAtari (Delfosse et al., 2024), and RealWorld RL Suite (Dulac-Arnold et al., 2020) that provide novelties to their respective base environments. Due to the dearth of established novelty detection tech-

niques in reinforcement learning, we compare our method to the one state-of-the-art RL novelty detection technique, RIQN (Danesh & Fern, 2021). We ablate our technique to more closely resemble standard assumptions about detection to show the possible dangers of finetuning a threshold fit for a novelty detector. While our primary result is built on top of the Dreamer world model (Hafner et al., 2021), we also show that our novelty bound technique can be applied to other types of world models with similar success.

## 2. Related Work

Novelty detection has taken on different names depending on the context (Pimentel et al., 2014). There are important applications that can benefit from RL novelty detection frameworks (Fu et al., 2017), yet novelty detection has not been well-studied in the realm of RL (Nasvytis et al., 2024).

Many complications arise from applying traditional machine learning novelty detection methods to an online setting (Müller et al., 2022). Generalization techniques exist, such as procedural generation to train agents to be more robust to novel situations (Cobbe et al., 2019) and data augmentation (Lee et al., 2019) to better train the agent, however these techniques suffer from high sample complexity (Müller et al., 2022). Furthermore, there may not be sufficient representation of the agent's evaluation environment to be able to detect novel Out-of-Distribution (OOD) transitions. Recent research shows that learning an OOD model is a difficult problem to solve and may even prove to be impossible in certain situations (Zhang et al., 2021; Fang et al., 2022). In addition, novelty detection techniques in the RL domain need to consider the current context of the situation. Not considering context becomes problematic when assuming that data is independently and identically distributed (i.i.d.), as it overlooks dependencies and contextual information that are often crucial for accurate detection. Thus it it not simple to apply standard detection techniques such as (Angiulli et al., 2024) in the RL domain.

Some methods attempt to use a measure of reward signal deterioration as a way to detect if a novelty has occurred, but that can prove to be dangerous in a high stakes environment where it may be too late to recover from a negative signal(Greenberg & Mannor, 2020). This method may also prove to have an incorrect mapping of the novelty that caused the reward shift.

The RIQN (Danesh & Fern, 2021) framework has shown to be successful by focusing on the aleoric uncertainty of the agent and draws upon the generated distributions of Implicit Quartille networks (Dabney et al., 2018). RIQN is an ensemble based method that detects novelty by predicting action values from state-action pairs as input, and predicts the feature distribution at each time step based on the previous values of the features. First, given an ensemble of $e$ dynamic models and time $t$, RIQN computes an anomaly score through predicting $e$ samples. Given these samples an anomaly score is computed through the average L1 distance for each observed feature between the samples and the actual observation at time $t$. The anomaly scores for each feature in the observation are then individually used as values for the cusum algorithm (Page, 1954) to detect any disturbances. However, RIQN is also sensitive to the selection of a threshold $\lambda$, and drift $\Delta$ which can vary from environment to environment and must be tuned; which requires *a priori* belief of whether anomalies will result in large or small shifts in aleoric uncertainty.

## 3. Background

**Partially observable Markov Decision Processes**  We study episodic Partially Observable Markov decision processes (POMDPs) denoted by the tuple $M = (\mathcal{S}, \mathcal{A}, \mathcal{T}, r, \Omega, O, \gamma)$, where $\mathcal{S}$ is the state space, $\mathcal{A}$ is the action space, $\mathcal{T}$ is the transition distribution $\mathcal{T}(s_t|s_{t-1}, a_{t-1})$, r is the reward function, $O$ is the observation space, $\Omega$ is an emissions model from ground truth states to observations, and $\gamma$ is the discounting factor (Åström, 1965).

**DreamerV2 World Model**  We conduct experiments applying our methods to a state-of-the-art DreamerV2 (Hafner et al., 2021) world model framework due to its VAE and history component, which is similar to traditional world model architectures (Ha & Schmidhuber, 2018a;b). DreamerV2 learns a policy by first learning a world model and then using the world model to roll out trials to train a policy model. The framework is composed of an image autoencoder and a Recurrent State-Space Model (RSSM). Relevant DreamerV2 components we will refer to throughout are:

- $x_t$ is the current image observation.
- $h_t$ is the encoded history of the agent.
- $z_t$ is an encoding of the current image $x_t$ that incorporates the learned dynamics of the world.

- $s_t = (h_t, z_t)$ is the agent's compact model state.

Given each representation of state $s_t$, DreamerV2 defines six other learned, conditionally-independent transition distributions given by the trained world model:

$$
\text{DreamerV2:}
\begin{cases}
\text{Recurrent model:} h_t = f_\phi(h_{t-1}, z_{t-1}, a_{t-1}) \\
\text{Representation model: } p_\phi(z_t|h_t, x_t) \\
\text{Transition prediction model: } p_\phi(\hat{z}_t|h_t) \\
\text{Image prediction model: } p_\phi(\hat{x}_t|h_t, z_t) \\
\text{Reward prediction model: } p_\phi(\hat{r}_t|h_t, z_t) \\
\text{Discount prediction model: } p_\phi(\hat{\gamma}_t|h_t, z_t)
\end{cases}
\tag{1}
$$

where $\phi$ describes the parameter vector for all distributions optimized. The loss function during training (Hafner et al., 2021) is:

$$
\mathcal{L}(\phi) = \mathbb{E}_{p_\phi(z|a,x)} \left[ \sum_t^T - \ln p_\phi(x_t|h_t, z_t) \right.
$$
$$
- \ln p_\phi(r_t|h_t, z_t)
$$
$$
- \ln p_\phi(\gamma_t|h_t, z_t)
$$
$$
\left. + \beta KL\left[p_\phi(z_t|h_t, x_t)||p_\phi(z_t|h_t)\right] \right]
\tag{2}
$$

where $\beta KL[p_\phi(z_t|h_t, x_t)||p_\phi(z_t|h_t)]$ is minimized by improving the prior dynamics towards the more informed posterior through KL Balancing (Hafner et al., 2021).

The goal of the DreamerV2 world model is to learn the dynamics and predictors for the observation $x_t$, reward, and discount factor of the training environment. We consider a single agent online RL setting, where an agent at time $t$ will traverse through each state $s_t$ in which observations are represented in the form of $x_t$. The agent relies on these along with $r_t$ to construct its belief state and make decisions to achieve the optimal discounted sum of rewards.

We consider an agent that has trained in a stationary training environment and then been deployed to a non-stationary evaluation environment that undergoes a change that is *a priori* unknown and unanticipated during agent development and training.

For latent-based detection, we investigate the world model and its learned probabilities. We construct a bound that is **not dependent on additional hyper-parameters** and test the world model's learned logic given the effect of the observation $x_t$ at time $t$. Our goal is to inform the agent in response to all possible Markovian novel observations. We do not consider techniques that are reliant on reward deterioration as those methods can involve temporally extended action sequences to be conducted between sparse rewards. Even with dense rewards, novelty may only be detected after a meaningful amount of time of reward decrease.

We also do not consider generalization techniques that try to model all possible novelties that the agent may experience, as it is generally impossible to construct a model that generalizes towards every unseen stimuli; to do so one must be able to categorize possible novelties in advance and generate novelties during training time (Wang et al., 2020).

The KL loss equation in (2) implies that, as training progresses, the divergence between the latent representation of the hidden state given the latent current state representation with and without the direct observation goes to zero for local transition space (Wang et al., 2019), i.e., $KL[p_\phi(z_t|h_t, x_t)||p_\phi(z_t|h_t)] \le \epsilon$. Given the current training mechanism in place, learning the transition distribution may be difficult due to the task of avoiding regularizing the representations toward a poorly trained prior (Hafner et al., 2021). A direct consequence of using common KL balancing techniques such as Equation (2) to bias the loss towards the prior error is that the world model is initially reliant on ground truth $x_t$ in contrast to a noisy $h_t$. This stabilizes $p_\phi(z_t|h_t, x_t)$ for proper training of $p_\phi(z_t|h_t)$ and $p_\phi(\hat{x}_t|h_t, z_t)$ (Asperti & Trentin, 2020) until the model understands the role of history $h_t$. Since the world model does not directly measure this relationship with $x_t$, we introduce the **cross entropy score comparison**:

$$
\begin{aligned}
& H(p_\phi(z_t|h_t, x_t), p_\phi(z_t|h_0)) \\
& - H(p_\phi(z_t|h_t, x_t), p_\phi(z_t|h_0, x_t))
\end{aligned} \tag{3}
$$

where $h_0$ is simply an empty hidden state passed to the model in order to simulate the dropout of $h_t$ and compute the influence that the ground truth $x_t$ has on the final prediction of the distribution of latent state $z_t$. This gives us an empirical measure of the world model's current reliance and improved performance based on the ground truth observation $x_t$, since the cross entropy score comparison *increases* if we find the entropy to be minimized with respect to $h_0$.

As the divergence between latent predictions with and without the ground truth observable decreases, $KL[p_\phi(z_t|h_t, x_t)||p_\phi(z_t|h_t)] \to \epsilon$, we expect a reduction of the impact of $x_t$. We model this relationship with the **novelty detection bound**:

$$
\begin{aligned}
KL[p_\phi(z_t|h_t, x_t)||p_\phi(z_t|h_t)] \le \\
KL[p_\phi(z_t|h_t, x_t)||p_\phi(z_t|h_0)] \\
- KL[p_\phi(z_t|h_t, x_t)||p_\phi(z_t|h_0, x_t)]
\end{aligned} \tag{4}
$$

where we substitute KL divergence in place of cross entropy loss to use the world model loss equation (2) (See Appx. D for a derivation).

The intuition behind the bound is as follows. If this relationship becomes disturbed given an unforeseen observation $\hat{x}_t$—that is, if the model is incapable of constructing a mapping such that the cross entropy of $H(p_\phi(z_t|h_t, x_t), p_\phi(z_t|h_0, x_t))$ is not minimized with respect to the performance of $H(p_\phi(z_t|h_t, x_t), p_\phi(z_t|h_0))$—the threshold (the right-hand side of the equality in (4)) will decrease. But it is also expected that the left-hand side, $KL[p_\phi(z_t|h_t, x_t)||p_\phi(z_t|h_t)]$, reaches extremely high values, depending on the current reliance of $h_t$.

Thus, there are two cases of detection arise upon seeing an observation $x_t$:

**Proposition 3.1.** *If the cross entropy score comparison becomes **negative** when introducing the vector $x_t$, then the right side of (4) will become negative, which immediately flags $x_t$ as novelty due to the property of non-negativity of the left side KL divergence.*

**Proposition 3.2.** *If the cross entropy score comparison becomes **nonnegative** when introducing the vector $x_t$, then 4 defines a decision boundary in the latent space over the measure of robustness to partial destruction of the input (Vincent et al., 2008; Srivastava et al., 2014), i.e.:*

$$
\begin{aligned}
KL[p_\phi(z_t|h_t, x_t)||p_\phi(z_t|h_t)] + \\
KL[p_\phi(z_t|h_t, x_t)||p_\phi(z_t|h_0, x_t)] \le \\
KL[p_\phi(z_t|h_t, x_t)||p_\phi(z_t|h_0)]
\end{aligned} \tag{5}
$$

where the model is tasked to simultaneously have sufficiently low KL divergence with the dropout of $x_t$ and with the dropout of $h_t$. Therefore, our bound is both combating posterior collapse, i.e., $p_\phi(z_t|x_t) = p_\phi(z_t|h_0)$, as well as measuring possible over-fitting i.e $KL[p_\phi(z_t|h_t, x_t)||p_\phi(z_t|h_0, x_t)] \gg c$ for some large $c$.

We find the above empirically to be true, as illustrated in Figure 1. As the agent trains in a stationary (no novelty) environment, the surprise (orange) (Itti & Baldi, 2005) settles and steadily decreases. Initially, the difference of KL divergences (blue) is below the orange line, and all new stimuli are flagged as novel, but the world model quickly learns that the environment is predictable. As the agent progresses through training, the green line represents how effective inputs $(h_0, x_t)$ are to the model in terms of predicting a distribution that minimizes $KL[p_\phi(z_t|h_t, x_t)||p_\phi(z_t|h_0, x_t)]$. Initially this value is close to the orange line because the model has learned to predict what will happen based mostly on the observation $x_t$. At some point the model discovers that incorporating history $h_t$ further drives loss decrease, and the KL divergence with the history dropout rises. A properly trained model is one in which its learned representations, $h_t$ and $x_t$, are useful in the prediction of distribution $p_\phi(z_t|h_t, x_t)$. Figure 1 illustrates that failure to do so may result in $x_t$ being interpreted as a noisy signal (blue differences collapse to orange) or the transition prediction model with $h_t$ has not yet met sufficiently low KL divergence (orange exceeds blue).

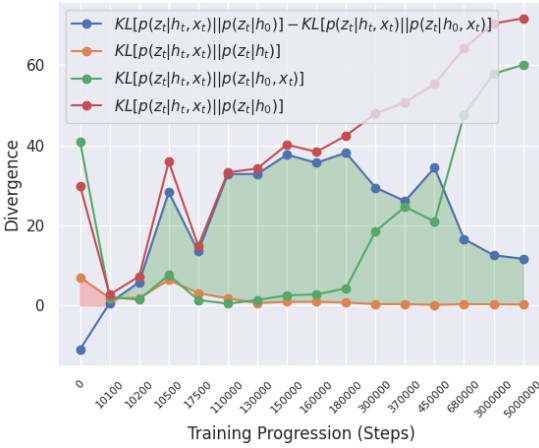

Figure 1: Visualization of the average levels of divergence given $x_t$ samples from the nominal MiniGrid (Chevalier-Boisvert et al., 2018) environment as training progresses: proposed bound (Blue); divergence of the RSSM predicted distributions given $(h_t)$ and $(h_t, x_t)$ (Orange); divergence between the RSSM given $(h_t, x_t)$ and the RSSM given only the image $(h_0, x_t)$ (Green line) in the Nominal environment; and the divergence of the RSSM given $(h_t, x_t)$ and receiving zero input $(h_0)$ (Red) as training progresses. For a training time-step $t$, a given $x_t$ is said to be a normal observation if the value of $KL[p_\phi(z_t|h_t, x_t)||p_\phi(z_t|h_t)]$ is below blue (within the green shaded area), otherwise $x_t$ is said to be a novel observation given the current context. See Appendix C.1 for similar visualizations corresponding to other environments.

## 4. Experiments

To empirically evaluate our bound, experiments are conducted in the MiniGrid (Chevalier-Boisvert et al., 2018), Atari (Machado et al., 2018), and continuous Deep Mind Control suite (DMC) (Tunyasuvunakool et al., 2020)[2]. We introduce novelties by, after a certain amount of time, generating novel observations from alternative environment configurations from NovGrid (Balloch et al., 2022), HackAtari (Delfosse et al., 2024) and the RealWorldRL Suite (Dulac-Arnold et al., 2020). We detail each of novel environments in Appendix E. We use the exact hyperparameters introduced in Hafner et al. (2021) for training DreamerV2, our base world model (c.f. our Appendix F).

We first train an agent to learn a $\epsilon$-optimal policy in the nominal environment then transfer the agent to one of the novel environments during testing time and let the trained agent take steps in each novel environment, capturing 300 independent and identically distributed episodes. We track each time step where novelty is detected to imitate possible agent halting situations. The agent's trained policy is expected to be sub-optimal in the novel environments but it is not re-trained at any point to adapt to the novel environments' dynamics. A novelty is experienced in every episode

---

[2]DMC experiments are conducted in pixel space, whereas most common usage is via position, velocity and orientation vectors.

traversed and all detection methods use the same policy.

### 4.1. Baselines

**RIQN** We compare our bound against the Recurrent Implicit Quantile Network anomaly detection model (RIQN) (Danesh & Fern, 2021), a traditional and accepted RL-focused novelty detection approach, taking note of some of the practical desiderata discussed in (Müller et al., 2022) to ground our evaluation techniques. We explicitly train the RIQN model using $10^6$ nominal transitions given by the trained policy as recommended (Danesh & Fern, 2021). We test the trained RIQN algorithm with the recommended hyper-parameters, as well as use the recommended cusum algorithm to detect novelties (Page, 1954), and the same trajectories used by the world model. In addition, we adjust the threshold $\lambda$, and drift $\Delta$ (value listed next to method name in tables) to improve the performance on larger observation dimensions. We use an ensemble size of 5 within the framework. RIQN represents the class of ensemble-based baselines when correcting for the RL setting.

**PP-Mare** We derive a novel ablation reconstruction error technique over the induced DreamerV2's RSSM *prior* and *posterior* reconstructions for a single realized step. This is a straightforward application of a world model wherein reconstructed states are directly compared; it is a simple, but effective, way to detect novelties and anomalies. However, it does require a tuned hyperparameter. We derive the method by first observing that locally successful training of the world model implies $\beta KL[p_\phi(z_t|x_t, h_t)||p_\phi(\hat{z}_t|h_t))] \leq \epsilon$. To properly utilize reconstruction error for trained world models, we induce $p_\phi(x_t|h_t, \hat{z}_t)$ from Eq. 2 where $h_t = f_\phi(h_{t-1}, z_{t-1}, a_{t-1})$, $z_{t-1} \sim p_\phi(z_{t-1}|h_{t-1}, x_{t-1})$, and $\hat{z}_t \sim p_\phi(z_t|h_t)$ to compare the $i$ pixel reconstruction losses between the generated 1-step prior and posterior $x_t$ observations and bound the difference by a small $\lambda$ defined by the user during deployment:

$$\frac{\sum_i^N |\hat{x}_{t\,\text{prior}}^i - \hat{x}_{t\,\text{posterior}}^i|}{N} \leq \lambda \qquad (6)$$

where $\hat{x}_{t\,\text{prior}} \sim p_\phi(\hat{x}_t|h_t, \hat{z}_t)$, and $\hat{x}_{t\,\text{posterior}} \sim p_\phi(\hat{x}_t|h_t, z_t)$. This removes direct dependence on the replay buffer by using only the final performance of the encoder and the current state $(z_t, h_t)$. As discussed further in Appendix A (see Figure 3 in the Appendix in particular), the prior and posterior observations can have dramatic changes in representation when the predicted latent representation $z_t$ is revealed.

We tune $\lambda$ (value listed next to method name in tables) to improve the performance of our PP-Mare baseline during all experiments. PP-Mare represents the class of observation-based baselines when correcting for the RL setting.

| Atari | Boxing | Kangaroo | Freeway | SeaQuest |
|---|---|---|---|---|
| KL Bound | $\leq 10^{-2}$ | $\leq 10^{-2}$ | $\leq 10^{-2}$ | $\leq 10^{-2}$ |
| PP-Mare | .04 | $\leq 10^{-2}$ | $\leq 10^{-2}$ | $\leq 10^{-2}$ |
| RIQN | .17 | .39 | .24 | .43 |

| DMC | Cartpole-B. | Quadruped-R. | Humanoid-S. | Walker-W. |
|---|---|---|---|---|
| KL Bound | $\leq 10^{-2}$ | $\leq 10^{-2}$ | $\leq 10^{-2}$ | $\leq 10^{-2}$ |
| PP-Mare | .01 | .01 | $\leq 10^{-2}$ | $\leq 10^{-2}$ |
| RIQN | .68 | .19 | .05 | .18 |

Table 2: False positive rates (lower is better) for nominal (no novelty) environments. PP-Mare and RIQN hyperparameters are tuned for each environment and correspond to those reported in Tables 3, 4, and 4 for each environment.

## 4.2. Metrics

We use the following metrics. *Novelty detection delay error* is the difference between the earliest time step that novelty is first observable and the time step that the novelty is detected. *Average delay error* (ADE) is the measure of how many steps off the detection method is from the true environment time step of the novelty. We calculate the agent's average delay error by averaging the novelty detection delay error across all novel environment episodes. *False positives*: We use the original nominal training environment to test for false positives. We do not consider false negatives when in the training environment. Ideal performance primarily minimizes the average delay error and false positive rates (Müller et al., 2022). In addition to average delay error, we also measure the *real time inference speed* to compare the raw speed of each method when computational resources are held fixed. Finally, we measure *AUC* to evaluate the discriminative ability of anomaly scores (left-hand side of Eqn. 4) generated by each method.

## 4.3. Results

**False Positives**   Table 2 shows false positive rates for Atari and DeepMind Control suite environments. The false positive rates for Minigrid-DoorKey-6x6 is $\leq 10^{-2}$, .03, and .52 for the KL bound method, PP-Mare, and RIQN.

The KL bound method false positives rate is $\leq 10^{-2}$ across all environments tested, allowing for minimal confrontation from a potential user. We hypothesize that KL's high performance in avoiding false positives coincides with the behavior observed from Figure 1 and is a direct result of the bounds formulation. RIQN's high false positive rate was also observed by Danesh & Fern (2021).

**Average Delay Error and AUC Scores**   Average Delay Error and AUC Scores are reported in Tables 3 (Atari), 4 (DeepMind Control Suite), and 5 (Minigrid). The KL bound appears to demonstrate a strong potential as a versatile approach for novelty detection, across all environments, particularly due to its ability to detect nuanced differences in anomaly score scales. Despite tuning $\lambda$ and $\Delta$ parameters

|  | ADE↓ | AUC↑ | ADE↓ | AUC↑ | ADE↓ | AUC↑ |
|---|---|---|---|---|---|---|
| **Boxing** | | OneArm | | BodySwitch | | Doppleganger |
| KL Bound | 52.6 | .708 | $\leq 10^{-2}$ | $\geq .99$ | $\leq 10^{-2}$ | $\geq .99$ |
| PP-Mare (2) | 22.5 | .605 | 6.30 | .915 | 5.4 | .862 |
| RIQN ($10^{-5}, 10^{-7}$) | 347.5 | .505 | 509.3 | .380 | 103.1 | .401 |
| **Kangaroo** | | Floorswap | | Difficulty+ | | DisableMonkey |
| KL Bound | 9.9 | .787 | $\leq 10^{-2}$ | $\geq .99$ | .960 | $\geq .99$ |
| PP-Mare (.5) | 85.2 | .281 | 42.5 | $\geq .99$ | 41.3 | .937 |
| RIQN ($10^{-2},10^{-2}$) | 166.3 | .541 | 94.2 | $\geq .99$ | 93.1 | .710 |
| **Freeway** | | InvisibleCars | | ColorCars | | FrozenCars |
| KL Bound | $\leq 10^{-2}$ | $\geq .99$ | $\leq 10^{-2}$ | $\geq .99$ | $\leq 10^{-2}$ | .985 |
| PP-Mare (.5) | 2.33 | $\geq .99$ | 1.84 | $\geq .99$ | 2.60 | .931 |
| RIQN ($10^{-7}, 10^{-9}$) | 2.87 | .938 | 2.62 | $\geq .99$ | .502 | .980 |
| **SeaQuest** | | DisableEnemy | | Gravity | | UnlimitedOxygen |
| KL Bound | .202 | .962 | 45.8 | .949 | $\leq 10^{-2}$ | $\geq .99$ |
| PP-Mare (.7) | 111.6 | .678 | 24.1 | .882 | 3.73 | .938 |
| RIQN ($10^{-2}, 10^{-3}$) | 45.3 | .272 | 157.3 | .390 | 16.0 | .701 |

Table 3: Average Delay Error and AUC results for **Atari** environments, with best tuned parameters when appropriate.

for RIQN, our proposed KL and PP-Mare methods achieved lower or comparable average delay error (ADE) and higher AUC to RIQN across all MiniGrid and Atari environments, while maintaining competitive performance in the DMC domain. RIQN's faster ADE scores in DMC may be achieved by its willingness to trade-off higher false positive rates in certain environments, such as Cartpole-3D-Balance where the false positive rate is as high as 68%, but never lower than 5% and usually somewhere in the middle. Arguably, high false-positive anomaly detection rates are undesirable. Additionally, RIQN exhibits greater sensitivity to subtle observation changes, as reflected in its strong performance in low-noise environments.

PP-Mare generally falls behind our KL bound, likely due to reconstruction error not being a measure optimized for computing semantic differences between the observations. Further, PP-Mare assumes a false correlation between pixels with high reconstruction error and novel regions of input images (Feeney & Hughes, 2021). In order to achieve a greater AUC for the PP-Mare method, a better similarity/difference metric would be needed to quantify the differences between the observations.

The AUC score also presents possible improvements that can be made over each method, as at times RIQN's cusum detection was unable to take advantage of decent AUC scores generated by each observation feature. The KL bound was effectively able to translate its higher AUC scores far more often than RIQN, despite no user interference.

**Real Time Halting Speed**   We analyze the real time halting speed to put ADE and FP rate into perspective for real world detection scenarios. Table 6 presents the real time halting speed increase in comparison to the RIQN algorithm.

|  | ADE↓ | AUC↑ | ADE↓ | AUC↑ | ADE↓ | AUC↑ |
|---|---|---|---|---|---|---|
| **Cartpole-3D-Balance** | LowPerterb | | HighPerterb | | LowNoise | |
| KL Bound | 51.9 | .799 | 49.2 | .774 | 24.1 | .853 |
| PP-Mare (.7) | 45.4 | .529 | 43.7 | .522 | 61.3 | .600 |
| RIQN ($10^{-1}$,$10^{-1}$) | 6.20 | .915 | 5.75 | .890 | 3.67 | $\geq$ .99 |
|  | HighNoise | | LowDamp | | HighDamp | |
| KL Bound | 2.86 | $\geq$ .99 | 20.2 | .784 | 25.0 | .770 |
| PP-Mare (0.7) | 22.7 | .840 | 110.8 | .395 | 94.5 | .448 |
| RIQN ($10^{-1}$,$10^{-1}$) | 3.33 | .923 | 5.33 | .543 | 1.66 | .520 |
| **Quadruped-3D-Run** | LowPerterb | | HighPerterb | | LowNoise | |
| KL Bound | 1.00 | .908 | 1.30 | .901 | 24.8 | .839 |
| PP-Mare (1.5) | 1.00 | .793 | 1.26 | .830 | 17.3 | .645 |
| RIQN ($10^{-1}$,$10^{-1}$) | 13.3 | .806 | 12.7 | .883 | 4.70 | .850 |
|  | HighNoise | | LowDamp | | HighDamp | |
| KL Bound | 3.82 | $\geq$ .99 | 89.8 | .512 | 112.9 | .507 |
| PP-Mare (1.5) | 3.94 | .821 | 49.1 | .448 | 52.86 | .415 |
| RIQN ($10^{-1}$,$10^{-1}$) | 4.02 | .980 | 94.3 | .426 | 103.0 | .489 |
| **Humanoid-3D-Stand** | LowPerterb | | HighPerterb | | LowNoise | |
| KL Bound | 44.4 | .925 | $\leq 10^{-2}$ | $\geq$ .99 | 9.05 | .755 |
| PP-Mare (4) | 17.9 | .833 | 15.2 | .849 | 157.0 | .363 |
| RIQN ($10^{-1}$,$10^{-1}$) | 64.8 | .854 | 32.3 | .983 | 2.16 | .919 |
|  | HighNoise | | LowFriction | | HighFriction | |
| KL Bound | $\leq 10^{-2}$ | $\geq$ .99 | 59.8 | .748 | 18.75 | .845 |
| PP-Mare (4) | 258.5 | .372 | 17.3 | .853 | 8.10 | .822 |
| RIQN ($10^{-1}$,$10^{-1}$) | 1.50 | .982 | 53.1 | .549 | 12.7 | .480 |
| **Walker-3D-Walk** | LowPerterb | | HighPerterb | | LowNoise | |
| KL Bound | $\leq 10^{-2}$ | $\geq$ .99 | $\leq 10^{-2}$ | $\geq$ .99 | 17.6 | .573 |
| PP-Mare (7) | 18.7 | .725 | 7.16 | .798 | 50.9 | .407 |
| RIQN ($10^{-1}$,$10^{-1}$) | 8.11 | .743 | 7.80 | .715 | 2.73 | .883 |
|  | HighNoise | | LowFriction | | HighFriction | |
| KL Bound | 1.48 | .911 | 17.25 | .961 | 5.93 | $\geq$ .99 |
| PP-Mare (7) | 21.1 | .546 | 2.37 | .943 | 1.98 | .962 |
| RIQN ($10^{-1}$,$10^{-1}$) | 2.10 | .986 | 10.55 | .700 | 13.6 | .722 |

Table 4: Average Delay Error and AUC results for **DeepMind Control** environments, with best tuned parameters when appropriate.

|  | ADE↓ | AUC↑ | ADE↓ | AUC↑ | ADE↓ | AUC↑ |
|---|---|---|---|---|---|---|
| **DoorKey-6x6** | LavaGap | | BrokenDoor | | DoorGone | |
| KL Bound | .110 | .732 | $\leq 10^{-2}$ | .939 | .120 | .940 |
| PP-Mare (1) | .170 | .765 | .017 | .784 | .080 | .685 |
| RIQN ($10^{-2}, 10^{-2}$) | $\leq 10^{-2}$ | .760 | 2.56 | .920 | .066 | .600 |
|  | Teleport | | ActionFlip | | | |
| KL Bound | $\leq 10^{-2}$ | .992 | $\leq 10^{-2}$ | .991 | | |
| PP-Mare (1) | .080 | .959 | .105 | .962 | | |
| RIQN ($10^{-2}, 10^{-2}$) | 6.29 | .950 | 2.34 | .890 | | |

Table 5: Average Delay Error and AUC results for **Minigrid** environments, with best tuned parameters when appropriate.

| Method | Average False Positive Rate | | Inference Run-time Speedup | |
|---|---|---|---|---|
|  | DMC | Atari | DMC | Atari |
| RIQN | .275 | .308 | $\times 1$ | $\times 1$ |
| PP-Mare | $\leq 10^{-2}$ | $\leq 10^{-2}$ | $\times 1.16 \cdot 10^2$ | $\times 4.45 \cdot 10^1$ |
| KL bound | $\leq 10^{-2}$ | $\leq 10^{-2}$ | $\times 1.34 \cdot 10^3$ | $\times 5.12 \cdot 10^2$ |

Table 6: Real time performance of RIQN versus our proposed RL-specific detection methods, in DeepMind Control Suite and the Arcade Learning Environment (Atari) under all respective tested novelties.

We observe that the KL computation increases the speed performance in both the Atari and DMC domains, by a magnitude of $10^2$ and $10^3$ respectively. Despite the heavier architecture of a world model, the advantage of using KL or PP-Mare over RIQN appears to be the removal of the need to compute an individual anomaly score over each $m$ feature (of the observation) for $e$ samples, as well as operating the cusum calculation over all $m$ features for each transition. Indeed, we see the increase in observation dimension between Atari and DMC vastly slows the RIQN algorithm as feature dimension begins to rise.

**Alternative World Models**    In this section we explore world model architectures other than the RNN-based DreamerV2 architecture. World models can alternatively be built on top of diffusion (Ho et al., 2020; Alonso et al., 2024), and transformer (Vaswani et al., 2017; Micheli et al., 2023) models. We provide detailed instructions (see Appendix B for details) required to replicate results on different world

model architectures, We report ADE, FP, and AUC in Table 7 for the `Atari-Freeway` environment.

The KL bound works with transformer-based world models and has very low false-positive rates as consistent with the bound on other architectures. The transformer-based IRIS (Micheli et al., 2023) is slower to build evidence of the novelty, which is consistent with observations by Micheli et al. (2023) (Appendix B) about under-sampled states on the `Atari-Freeway` environment; the effect on novelty detection is described further in our Appendix C.2. Our KL bound method cannot be used with diffusion-based world-models that directly generate prior and posterior images instead of latent hidden states. Instead, we use PP-Mare, the closest ablation that operates directly on predicted observation differences.

**Exploration Effects**    We turn our attention toward the hypothetical KL divergence of:

$$KL[p(z_t|h_t, x_t) \parallel p_\phi(z_t|h_t, x_t)] \leq \epsilon$$

where $p(z_t|h_t, x_t)$ is the theoretical true distribution of the training environment.

If a world model is trained alongside a policy, the world model is restricted to train on the local transition space as the agent zeros in on particular trajectories of states with high value as the policy improves (Kauvar et al., 2023). This can result in the phenomenon where the policy restricts what the world model is able to observe and learn from. In these cases, aspects of the environment that have not been observed enough may appear novel even. This is

| | ADE↓ | AUC↑ | ADE↓ | AUC↑ | ADE↓ | AUC↑ | FP↓ |
|---|---|---|---|---|---|---|---|
| **Atari Freeway** | InvisibleCars | | ColorCars | | FrozenCars | | |
| RNN (DreamerV2) | $\leq 10^{-2}$ | $\geq .99$ | $\leq 10^{-2}$ | $\geq .99$ | $\leq 10^{-2}$ | .985 | $\leq 10^{-2}$ |
| Diffusion-based | $\leq 10^{-2}$ | .978 | $\leq 10^{-2}$ | .977 | $\leq 10^{-2}$ | .988 | $\leq 10^{-2}$ |
| Transformer-based | 8.35 | .771 | 2.83 | .876 | 13.4 | .681 | $\leq 10^{-2}$ |

Table 7: We compare our KL-bound results implemented on RNN-based DreamerV2 (Hafner et al., 2021) to alternative world model architectures: diffusion-based (Alonso et al., 2024) and transformer-based (Micheli et al., 2023).

Table 8: Metrics for Fake Goal Environment

| Method | Accuracy |
|---|---|
| KL | 0.49 |
| PP-Mare | 0.46 |
| RIQN | 0.21 |
| KLExplored | 0.91 |

technically correct because the agent is observing aspects of the environment that it failed to learn about during training. However, it is also not always what is intended because the novelty detection is triggering off of the wrong aspects.

To give a concrete example, in `MiniGrid-DoorKey-6x6`, an agent that is trained to always go from the left-hand side of the map to a goal on the right-hand side of the map will never need to turn to look at the back side of a door. However, if the novelty is that it starts on the right side of the door, it will see the back side of the door, as in Figure 2. This transition was possible during the training of the world model, but it was illogical to consistently explore looking backwards and the policy quickly learned to discourage that behavior, depriving the world model of observations that should have been able to occur naturally. After the novelty, the world model's locally learned dynamics classifies this transition as a novelty.

To illustrate that it is an artifact of how the agent is learned, we provide an alternative training paradigm, *KLExplored* where we train the agent from both sides of the map pre-novelty. Table 8 shows that this increases novelty detection accuracy because this transition is no longer flagged. We leave the decision of declaring a novel transition as novel based on the local environment dynamics or the agent's beliefs for an entirely different discussion (Miljković, 2010; Balloch et al., 2022). This highlights the significance that novelties are unanticipatable because a training paradigm that is prepared for this case will be more accurate. However, we assume that novelties cannot be anticipated in advance and thus factored into the training paradigm.

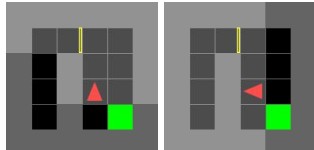

Figure 2: Minigrid full render of a simple `FakeGoal` environment. Here we empirically observe when the agent detects novelty in an task that has already been completed by disabling the goal. For this ablation experiment, we tentatively define ground truth novel transitions as transitions that interact with the fake goal. The most common transition initially flagged is from left to right. We suspect that this is due to the observation (light gray) that there appears to be nothing on the other side of an open door.

## 5. Limitations

Experimentation with novelty detection algorithms can be difficult as it requires environments that have novel alternative environments. To detect a novelty, one must be able to alter the dynamics of the environment. This creates a conundrum where experimentally we know the alternatives but must not allow data about the alternatives to inform the training of agents in the nominal, pre-novelty environment. Experimenter bias is also a risk faced by those that research novelty detection—our PP-Mare and KL-Bound methods were developed before we identified Atari and DeepMind Control Suite as sources of environments with alternatives.

While the research challenge of novelty detection is inspired by the real world, where sudden, unanticipatable, and permanent changes to how the world works occur relatively frequently, experimentation in the real world is often infeasible, requiring virtual surrogate testbeds.

While our proposition using KL divergence has proven effective empirically, future work might explore alternative divergence measures with metric properties, such as the Jensen-Shannon divergence or Wasserstein distance, which could provide stronger theoretical guarantees. Additionally, exploring the relationship between latent space dimensionality and the reliability of variational approximations (of which our bound is based on see Appendix D) could yield insights into optimizing model architecture for novelty detection.

## 6. Conclusions

Novelties are sudden changes to the observation space or environment state transition dynamics that occur at inference time that are unanticipated (or unanticipatable) by the agent during training. Novelties are permanent distribution shifts, distinguishing them from anomalies that are localized one-time out-of-domain occurrences. Novelties happen frequently in the real world and this paper addresses the detection of novelties, as defined above, but leaves the re-

sponse to novelties as out of scope.

This paper proposes a novel way to detect novelty in reinforcement learning settings using a world model. The KL bound method we introduce is demonstrably resilient to false positives while simultaneously detecting novelty quickly and accurately. Crucially, our KL bound method does not require thresholds or other hyperparameters. This is essential because tuning thresholds and hyperparameters requires some *a priori* intuition about the nature, scope, and scale of anomalies, which is an assumption that is disallowed in our research setting on novelty.

Our paper supports the overall value of world model based RL implementations because the world model can be repurposed to anomaly and novelty detection. Our method appears robust enough to provided the basis for future research on addressing inference-time novelties such as those that occur in the real world.

## Acknowledgments

We gratefully acknowledge Minh Vu, Manish Bhattarai, and Sebastián Gutiérrez Hernández for their valuable insights and stimulating discussions, which significantly contributed to the development and refinement of the ideas presented in this work. Their thoughtful feedback and engagement were instrumental in helping shape certain areas.

## Impact Statement

This work advances the field of reinforcement learning by exploring methods for detecting unanticipated changes in the environment, with a deliberate emphasis on minimizing reliance on additional hyperparameters. By leveraging world models in conjunction with a novel KL bound technique, we present a reliable and efficient approach to identifying persistent distributional shifts. Our findings highlight the broader utility of world models beyond policy learning, demonstrating their potential for robust deployment in dynamic, real-world settings where adaptability and resilience are essential. This contribution establishes a foundation for future systems capable of not only detecting but also responding to novelty with minimal human oversight.

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

# A. Reconstruction Error Comparisons

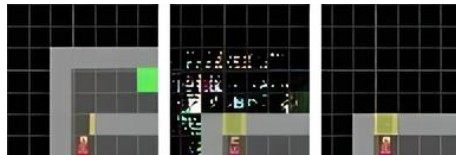

Figure 3: From left to right, $\hat{x}_{t_{prior}}, \hat{x}_{t_{posterior}}$ and $x_t$ observations of agent trying to open door in the `BrokenDoor` custom minigrid environment where the door fails to open despite having the correct key. The intuition of PP-Mare is to distinguish high reconstruction loss between samples.

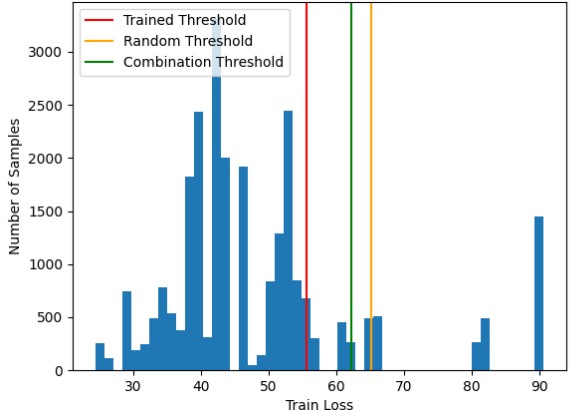 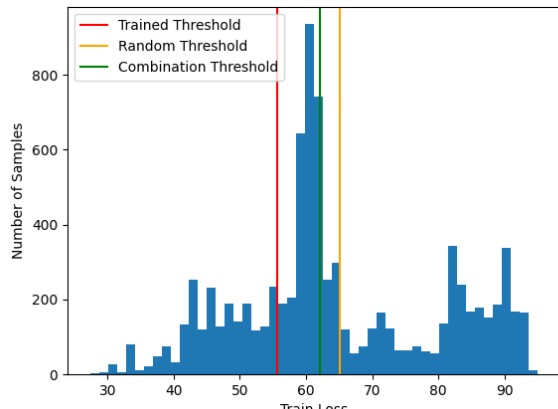

Figure 4: Reconstruction error alongside the three tested thresholds: Random Model, Trained Model and Combination model. Reconstruction error from the nominal `MiniGrid-DoorKey-6x6` environment (Left), and Reconstruction error novel `LavaGap` environment (Right). Each vertical line refers to the cut off value for the corresponding threshold. Samples to the right of the line are classified as novelty. Reconstruction Error was generated from a trained agent. It appears that no threshold (even tuned) would separate the space.

# B. Novelty Detection For Other World Model Frameworks

We consider world models architectures that learn representations of past and present states, alongside a distinct predictive model of the future distribution of states (Werbos, 1987), preferably a powerful predictive model implemented on a general purpose computer such as a recurrent neural network (RNN) (Schmidhuber, 1990) (Ha & Schmidhuber, 2018a; Hafner et al., 2021). We show how our technique can be used on a RNN-based, Transformer-based, and Diffusion-based World Model respectively:

- **IRIS:** (Micheli et al., 2023) At a high level, the Transformer G captures the environment dynamics by modeling the language of the discrete autoencoder over time. We expect that the categorical probability distribution prompted from the predicted observation logits when generating $z_t$ can be used to compute the divergence between two outputs. Since the two world models are sufficiently different we briefly experiment with a possible interpretation of KL to the IRIS framework. We first construct the bound similarly to Eq 4. We model $p_\phi(z_t|h_0, x_t)$ as an auto-regressive prediction of the logits $z_t^0$ to $z_t^k$ where $z_t^0$ is predicted from the true observation $x_t$. We disable kv caching to mimic $h_0$. We model $p_\phi(z_t|h_0)$ as an auto-regressive prediction of the logits $z_t^0$ to $z_t^k$ where $z_t^0$ is predicted from the zero vector substituting for $x_t$ in the IRIS framework, we also disable kv caching to mimic $h_0$. We experiment with interpreting $p_\phi(z_t|h_t, x_t)$ as $p_\phi(z_t|h_0, x_t)$ except $p_\phi(z_t|h_t, x_t)$ utilizes kv caching. For our results, we experimented with bounding by $KL[p_\phi(z_t|h_t, x_t)||p_\phi(z_t|h_0)] - KL[p_\phi(z_t|h_t, x_t)||p_\phi(z_t|h_0, x_t)]$. We interpret the Bayesian surprise as $KL[p_\phi(z_t|z_{\leq t}, a_{\leq t})||p_\phi(z_t|\hat{z}_{t-1}, z_{\leq t-2}, a_{\leq t})]$, where $\hat{z}_{t-1}$ is generated from a previous sampling of $p_\phi(z_t|h_t, x_t)$. For our results, we experimented with utilizing $KL[p_\phi(z_t|z_{\leq t}, a_{\leq t})||p_\phi(z_t|\hat{z}_{t-1}, z_{\leq t-2}, a_{\leq t})]$ as a score for each auto-regressive step. Although our initial interpretation had moderate success, we expect that future work should be able to improve by adjusting the bound as well as addressing the exploration concerns given the exploration trick used in (Micheli et al., 2023), of which the "up" action was heavily sampled in favor of helping the agent complete the game,

rather than improving the world model performance. Ultimately, we leave the task of enhancing novelty detection in the IRIS world model for future work.

- **Diamond:** (Alonso et al., 2024) constructs a conditional generative model of the environment dynamics, $p_\phi(x_{t+1}|x_{\leq t}, a_{\leq t})$, and considers the general case of a POMDP, in which the Markovian state $s_t$ is unknown and can be approximated from past observations and actions. To construct $p$, the authors condition a diffusion model on the history, to estimate and generate the next observation directly. Since Diamond operates primarily in the observation space, we use PP-Mare to employ a interpretation of our technique. We utilize $p_\phi(x_{t+1}|x_{\leq t}, a_{\leq t})$ as our prior, and $x \sim p_\phi(x_{t+1}|x_{\leq t+1}, a_{\leq t+1}) = p(x_{t+1}|s_{t+1})$ as our posterior at each step.

## C. Policy Exploration and Uncertainty Effects On Detection

### C.1. Base Divergences in the Nominal Environment as Training Progresses

Figure 5 shows how the KL bound evolves with training for Atari environments. Each row is equivalent to Figure 1 but with each component of the bound split into a separate graph:

- $KL[p_\phi(z_t|h_t, x_t)||p_\phi(z_t|h_0)] - KL[p_\phi(z_t|h_t, x_t)||p_\phi(z_t|h_0, x_t)]$, right-hand side of Eqn 4.

- $KL[p_\phi(z_t|h_t, x_t)||p_\phi(z_t|h_t)]$, left-hand side of Eqn 4

- $KL[p_\phi(z_t|h_t, x_t)||p_\phi(z_t|h_0, x_t)]$, the subtracted component of the left-hand side of Eqn 4

- $KL[p_\phi(z_t|h_t, x_t)||p_\phi(z_t|h_0)]$, the component of Eqn 4 that is subtracted against.

Note that the scale of the $y$-axes differ. The significant observation is that in the nominal environment, the right-hand side of the bound is always higher than the left-hand side of the bound.

### C.2. Exploration effects

We turn our attention toward the hypothetical KL divergence of: $KL[p(z_t|h_t, x_t)||p_\phi(z_t|h_t, x_t)] \leq \epsilon$ where $p(z_t|h_t, x_t)$ is the theoretical true distribution of the training environment. If a world model is trained alongside a policy, the world model is restricted to train on the local transition space as the agent zeros in on particular trajectories of states with high value as the policy improves (Kauvar et al., 2023).

This can result in the phenomenon where the policy restricts what the world model is able to observe and learn from. In these cases, aspects of the environment that have not been observed enough may appear novel even. This is technically correct because the agent is observing aspects of the environment that it failed to learn about during training. However, it is also not always what is intended because the novelty detection is triggering off of the wrong aspects.

To give a concrete example, in `MiniGrid-DoorKey-6x6`, an agent that is trained to always go from the left hand side of the map to a goal on the right hand side of the map will never have need to turn to look at the back side of a door. However, if the novelty is that it starts on the right side of the door, it will see the back side of the door, as in Figure 2. This transition was possible during the training of the world model, but it was illogical to consistently explore looking backwards and the policy quickly learned to discourage that behavior, depriving the world model of observations that should have been able to occur naturally. After the novelty, the world model's locally learned dynamics classifies this transition as a novelty. To illustrate that it is an artifact of how the agent is learned, we provide an alternative training paradigm, *KLExplored* where we train the agent from both sides of the map pre-novelty. Table 8 shows that this increases novelty detection accuracy because this transition is no longer flagged. We leave the decision of declaring a novel transition as novel based on the local environment dynamics or the agent's beliefs for an entirely different discussion (Miljković, 2010; Balloch et al., 2022). This highlights the significance that novelties are unanticipatable because a training paradigm that is prepared for this case will be more accurate. However, we assume that novelties cannot be anticipated in advance and thus factored into the training paradigm.

### C.3. Policy Behavior against KL Divergence

**World Model vs Agent Uncertainty** Since the policy is configured to make decisions solely on the world model's predicted belief states $(z_t, h_t)$, in Figure 6 we analyze the effect of the world model on the policy's entropy (the degree of

**Atari-Boxing**

**Atari-Seaquest**

**Atari-Kangaroo**

**Atari-Freeway**

Figure 5: Divergences during training (Note the scale of the y axis)

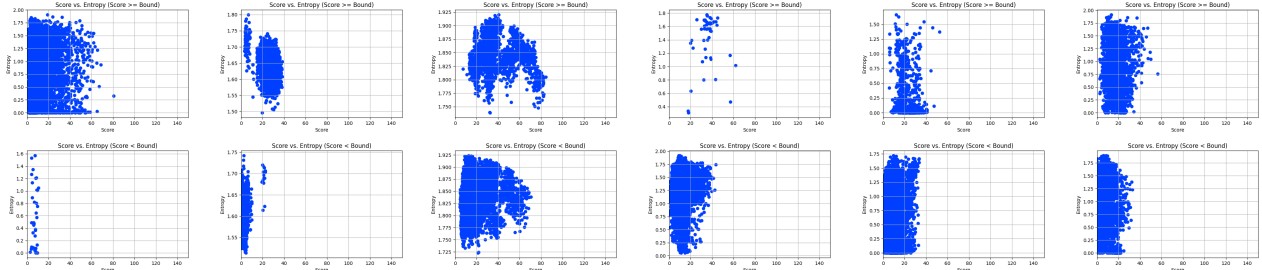

Figure 6: Novelty score (x-axis) against the entropy of the policy (y-axis) during different stages of training. Here we show that regardless of novelty being predicted (Top) or not (Bottom), the entropy of the policy does not appear to be strongly affected by the divergence of the posterior and prior, regardless of current training performance.

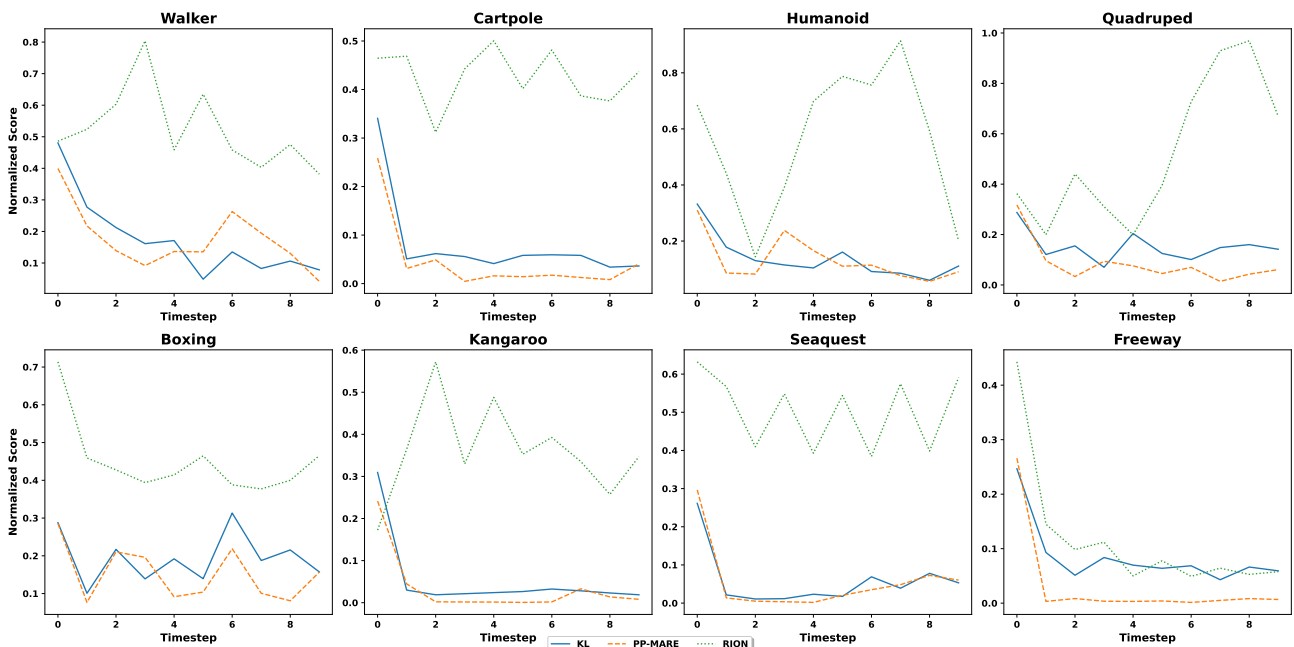

Figure 7: The normalized anomaly score trend of each detection method as the episode progresses 50 time-steps in the nominal environment.

disorder or uncertainty in a system) during the case of novelty detection and vice versa. From our experiments, it appears that regardless of the measured surprise from the world model, the policies entropy maintains a weak relationship with the world model throughout the entirety of training. This suggests that simply relying on the policy may be powerless in determining if a novel transition has occurred.

**Detection Score Trend Over Time** Figure 7 presents the normalized anomaly scores, (the left-hand side of Eqn 4 for KL, and Eqn 6 for PP-MARE) for each environment in the Atari and DMC domains as timestep progresses in the nominal, novelty-free environments. KL Bound and PP-Mare detection methods are expected to have higher anomaly scores in the first few steps of a training episode because the world model struggles to generate reasonable predictions due to its arbitrary prior initialization. Anomaly detection drops rapidly or was never high to begin with. This is in contrast to RIQN, where the anomaly score fluctuates and is higher for all environments except one. It appears that leveraging learned surrogates conditioned on the agent's actions appears to provide more informative signals of normality in higher-dimensional spaces, enabling the detection of novelties with minimal delay error.

# D. Derivations

**Mapping from cross entropy to KL divergence:**

$$H(p_\phi(z_t|h_t, x_t), p_\phi(z_t|h_0)) - H(p_\phi(z_t|h_t, x_t), p_\phi(z_t|h_0, x_t)) =$$
$$(KL[p_\phi(z_t|h_t, x_t)||p_\phi(z_t|h_0)] + H(p_\phi(z_t|h_t, x_t)))$$
$$-KL[p_\phi(z_t|h_t, x_t)||p_\phi(z_t|h_0, x_t)] - H(p_\phi(z_t|h_t, x_t)) =$$
$$KL[p_\phi(z_t|h_t, x_t)||p_\phi(z_t|h_0)]$$
$$-KL[p_\phi(z_t|h_t, x_t)||p_\phi(z_t|h_0, x_t)]$$

**Proposition 3.1 (restated)**    *If the cross entropy score comparison becomes **negative** when introducing the vector $x_t$, then the right side of (4) will become negative, which immediately flags $x_t$ as novelty due to the property of non-negativity of the left side KL divergence.*

$$H(p_\phi(z_t|h_t, x_t), p_\phi(z_t|h_0)) - H(p_\phi(z_t|h_t, x_t), p_\phi(z_t|h_0, x_t)) < 0 \implies$$
$$KL[p_\phi(z_t|h_t, x_t)||p_\phi(z_t|h_0)] - KL[p_\phi(z_t|h_t, x_t)||p_\phi(z_t|h_0, x_t)] < 0. \text{ Then,}$$
$$KL[p_\phi(z_t|h_t, x_t)||p_\phi(z_t|h_t)] < KL[p_\phi(z_t|h_t, x_t)||p_\phi(z_t|h_0)] - KL[p_\phi(z_t|h_t, x_t)||p_\phi(z_t|h_0, x_t)] \implies$$

$KL[p_\phi(z_t|h_t, x_t)||p_\phi(z_t|h_t)] < 0$; Which is impossible.

**Proposition 3.2 (restated)**    *If the cross entropy score comparison becomes **nonnegative** when introducing the vector $x_t$, then 4 defines a decision boundary in the latent space over the measure of robustness to partial destruction of the input (Vincent et al., 2008; Srivastava et al., 2014), i.e:*

$$KL[p_\phi(z_t|h_t, x_t)||p_\phi(z_t|h_t)] + KL[p_\phi(z_t|h_t, x_t)||p_\phi(z_t|h_0, x_t)] \leq$$
$$KL[p_\phi(z_t|h_t, x_t)||p_\phi(z_t|h_0)]$$

$H(p_\phi(z_t|h_t, x_t), p_\phi(z_t|h_0)) - H(p_\phi(z_t|h_t, x_t), p_\phi(z_t|h_0, x_t)) > 0$:

$$H(p_\phi(z_t|h_t, x_t), p_\phi(z_t|h_0)) - H(p_\phi(z_t|h_t, x_t), p_\phi(z_t|h_0, x_t)) \geq 0 \implies$$
$$KL[p_\phi(z_t|h_t, x_t)||p_\phi(z_t|h_0)] - KL[p_\phi(z_t|h_t, x_t)||p_\phi(z_t|h_0, x_t)] \geq 0 \implies$$
$$KL[p_\phi(z_t|h_t, x_t)||p_\phi(z_t|h_0)] - KL[p_\phi(z_t|h_t, x_t)||p_\phi(z_t|h_0, x_t)] \geq \epsilon \implies$$
$$KL[p_\phi(z_t|h_t, x_t)||p_\phi(z_t|h_0)] \geq \epsilon + KL[p_\phi(z_t|h_t, x_t)||p_\phi(z_t|h_0, x_t)] \implies$$
$$KL[p_\phi(z_t|h_t, x_t)||p_\phi(z_t|h_0)] \geq KL[p_\phi(z_t|h_t, x_t)||p_\phi(z_t|h_t)]^* + KL[p_\phi(z_t|h_t, x_t)||p_\phi(z_t|h_0, x_t)]$$
$$\text{(Iff the local training of } KL[p_\phi(z_t|h_t, x_t)||p_\phi(z_t|h_t)] \leq \epsilon \text{ holds for some } \epsilon)^*$$

$$(7)$$

Let the full posterior $p_\phi(z_t|x_t, h_t)$ be the most informed distribution and the desired distribution with support over the latent space Z at any given $t$, and let $f_\theta$ map to the representation of a distribution with the support of Z as well.

Utilizing notation similar to that in Vincent et al. (2008), denote $(h_t, x_t)$ as the clean input $x$ (not to be confused with $x_t$), and define three variants of $x'$: $\{x'_{(-h_t)}, x'_{(-x_t)}, x'_{(-h_t, -x_t)}\}$, which denote the removal of features $(h_t)$, $(x_t)$, and $(h_t, x_t)$ from the clean input $x$, respectively.

Since the loss is at the latent level, the robustness to partial destruction of a single input $x$ for a desired distribution is measured as:

$$L(p_\phi(z_t|h_t, x_t), f_\theta(x'))$$

Therefore, if $L$ is chosen to be the KL divergence, and $x'$ becomes explicit, then Proposition 3.2 as can be rewritten as:

$$L(p_\phi(z_t|h_t, x_t), p_\phi(z_t|h_0)) > L(p_\phi(z_t|h_t, x_t), p_\phi(z_t|h_t))^* + L(p_\phi(z_t|h_t, x_t), p_\phi(z_t|h_0, x_t))$$

$$KL(p_\phi(z_t|h_t, x_t) \| p_\phi(z_t|h_0)) > KL(p_\phi(z_t|h_t, x_t) \| p_\phi(z_t|h_t))^* + KL(p_\phi(z_t|h_t, x_t) \| p_\phi(z_t|h_0, x_t))$$

where $p_\phi$ corresponds to the distribution modeled by $f_\theta$ given some $x'$, and $h_0$ represents the state of $h$ when no conditioning information is available.

**Theoretical Guarantee of Equation 4**   **Assumption:** Suppose the objective defined by Equation 2 is minimized such that:

$$\beta\, KL\left[p_\phi(z_t|h_t, x_t) \middle\| p_\phi(z_t|h_t)\right] = 0,$$

implying that $z_t$ is conditionally independent of $x_t$ given $h_t$. Furthermore, assume that the model distribution $p_\theta$ is correctly specified, meaning there exists a set of parameters $\theta^*$ such that:

$$p_\phi(\cdot|\cdot) = p^*(\cdot|\cdot),$$

where $p^*(\cdot|\cdot)$ denotes the well defined true conditional distribution. Then we can rewrite Proposition 3.2 as:

$$KL(p_\phi(z_t|h_t, x_t) \,\|\, p_\phi(z_t|h_0)) > KL(p_\phi(z_t|h_t, x_t) \,\|\, p_\phi(z_t|h_0, x_t))$$

Note that the explicit definition is:

$$\mathrm{KL}\left(p_\phi(z_t \mid h_t, x_t) \middle\| p_\phi(z_t \mid h_0)\right) = \mathbb{E}_{p_\phi(z_t|h_t,x_t)}\left[\log \frac{p_\phi(z_t \mid h_t, x_t)}{p_\phi(z_t \mid h_0)}\right],$$

$$\mathrm{KL}\left(p_\phi(z_t \mid h_t, x_t) \middle\| p_\phi(z_t \mid h_0, x_t)\right) = \mathbb{E}_{p_\phi(z_t|h_t,x_t)}\left[\log \frac{p_\phi(z_t \mid h_t, x_t)}{p_\phi(z_t \mid h_0, x_t)}\right].$$

Now consider the difference:

$$\Delta = \mathrm{KL}\left(p_\phi(z_t \mid h_t, x_t) \middle\| p_\phi(z_t \mid h_0)\right) - \mathrm{KL}\left(p_\phi(z_t \mid h_t, x_t) \middle\| p_\phi(z_t \mid h_0, x_t)\right).$$

$$\begin{aligned}
\Delta &= \mathbb{E}_{p_\phi(z_t|h_t,x_t)}\left[\log \frac{p_\phi(z_t \mid h_t, x_t)}{p_\phi(z_t \mid h_0)}\right] - \mathbb{E}_{p_\phi(z_t|h_t,x_t)}\left[\log \frac{p_\phi(z_t \mid h_t, x_t)}{p_\phi(z_t \mid h_0, x_t)}\right] \\
&= \mathbb{E}_{p_\phi(z_t|h_t,x_t)}\left[\log \frac{p_\phi(z_t \mid h_t, x_t)}{p_\phi(z_t \mid h_0)} - \log \frac{p_\phi(z_t \mid h_t, x_t)}{p_\phi(z_t \mid h_0, x_t)}\right] \\
&= \mathbb{E}_{p_\phi(z_t|h_t,x_t)}\left[\log \frac{p_\phi(z_t \mid h_0, x_t)}{p_\phi(z_t \mid h_0)}\right].
\end{aligned}$$

Thus, we have a key result:

$$\Delta = \mathbb{E}_{p_\phi(z_t|h_t,x_t)}\left[\log \frac{p_\phi(z_t \mid h_0, x_t)}{p_\phi(z_t \mid h_0)}\right].$$

The inequality

$$\mathrm{KL}\left(p_\phi(z_t \mid h_t, x_t) \middle\| p_\phi(z_t \mid h_0)\right) \geq \mathrm{KL}\left(p_\phi(z_t \mid h_t, x_t) \middle\| p_\phi(z_t \mid h_0, x_t)\right)$$

thus holds whenever

$$\mathbb{E}_{p_\phi(z_t|h_t,x_t)}\left[\log \frac{p_\phi(z_t \mid h_0, x_t)}{p_\phi(z_t \mid h_0)}\right] \geq 0.$$

Which can be interpreted as holding iff the Expected Information Gain (EIG) of $x_t$ is nonnegative under the distribution of $p_\phi(z_t \mid h_t, x_t)$.

**Discussion on KL Divergence Properties**   We briefly note that KL divergence is not a true metric. The bound in Proposition 3.2 compares the posterior to three different priors and provides a useful approximation for detection, rather than a strict mathematical guarantee. Formally, for distributions P, Q, and R, the KL divergence does not generally satisfy $D_{KL}(P \,\|\, R) \leq D_{KL}(P \,\|\, Q) + D_{KL}(Q \,\|\, R)$. This means our bound might be violated not only due to novelty but potentially due to the inherent non-metric behavior of KL divergence, particularly in high-dimensional or complex latent spaces.

# E. Novel Environments

Below we state each novel environment alongside what timestep $T$ was used to evaluate our method (Note that frame number is different from $T$):

- **MiniGrid**
  - DoorKey-6x6
    * `BrokenDoor`: The door no longer opens, even with the key. T is set to the timestep when the agent attempts to open the door.
    * `ActionFlip`: All movement actions are set to the opposite direction. T is set to when the agent chooses to rotate.
    * `Teleport`: Teleport the agent randomly then perform the selected action. T is set to 5.
    * `HeavyKey`: The key can no longer be picked up. T is set to when the agent tries to pick up the key.
    * `LavaGap`: A lava gap is introduced around the goal. T is set to when the lava is in the agent's sight.
    * `DoorGone`: The door is removed from the environment and is replaced with empty space. T is set when the empty space is in the agent's sight.

- **Atari**
  - Freeway
    * `InvisibleCars`: All cars are invisible but still move. T = 100.
    * `ColorCars`: All cars are changed to black. T=100.
    * `FrozenCars`: All cars are suddenly frozen, reset, and do not move. T=100.
  - Kangaroo
    * `FloorSwap`: The agent suddenly switches onto a different floor. T = 300.
    * `Difficulty+`: The games difficulty switches to the hardest. T = 300.
    * `DisableMonkey`: All monkeys are removed from the game. T is set to when the monkeys are fully on the screen.
  - SeaQuest
    * `DisableEnemy`: Enemies are removed from the game. T is set when the enemies are fully on the screen. T = 300.
    * `UnlimitedOxygen`: The player suddenly gains unlimited oxygen. T = 500.
    * `GravityDrift`: The gravity begins to become stronger. T = 300.
  - Boxing
    * `OneArm`: One of the agent's arms are disabled. T is set to the time-step when the agent first attempts to use the disabled arm.
    * `BodySwitch`: The agent and the opponent switch places visually. T = 300.
    * `Doppleganger`: The agent and opponent look exactly the same. T = 300.

- **DeepMind Control Suite**
  - All environments use the following:
    * `Perturb`: The agent suddenly has a limb length increase. T = 1.
      · Walker: Thigh length increase to .3 (Low) or 1 (High).
      · Humanoid: Head size increase to .2 (Low) or .3 (High).
      · Cartpole: Pole mass increase to 5 (low) or 10 (High).
      · Quadruped: Shin length increase to 1 (low) or 2 (High).
    * `Noise`: The agent's observations begin to be filled with gaussian noise. Standard Deviation: 5 (Low) or Standard deviation: 30 (High). T = 30.
    * `Friction`: The agent's environment has 6 (Low) or 10 (High) increased friction. T = 1.
    * `Damping`: The agent's joints have 1 (Low) or 2 (High) increased damping. T = 30.

We first train an agent to learn a $\epsilon$-optimal policy in the nominal environment, then transfer the agent to one of the environments with novelties during testing time. We then let the trained agent take steps in each novel environment, capturing 50,000 steps within various independent and identically distributed episodes; tracking each time step where novelty is detected to imitate possible agent halting situations. The agent's trained policy is expected to be sub-optimal in the novel environments and is not re-trained at any point to adapt to the novel environments' dynamics. All novelties are experienced in every episode traversed.

## F. Hardware Requirements

All experiments can be sufficiently reproduced utilizing a NVIDIA GeForce GTX 1080 GPU with at least 8 GB of VRAM for environment complexity, a AMD Ryzen 5 5600X 6-Core Processor and at least 50 MB for files, excluding training data which is dependent on environment and model hyper-parameters.

Future work would likely go beyond the scope of these hardware requirements, and we expect that cloud computing is a necessity to experiment in larger domains.

### F.1. Dreamer World Model Training Parameters

| Name | Symbol | Value |
|---|---|---|
| World Model | | |
| Dataset size (FIFO) | — | $2 \cdot 10^6$ |
| Batch size | $B$ | 50 |
| Sequence length | $L$ | 50 |
| Discrete latent dimensions | — | 32 |
| Discrete latent classes | — | 32 |
| RSSM number of units | — | 1024 |
| KL loss scale | $\beta$ | 1 |
| KL balancing | $\alpha$ | 0.8 |
| World model learning rate | — | $2 \cdot 10^{-4}$ |
| Reward transformation | — | $\tanh$ |
| Behavior | | |
| Imagination horizon | $H$ | 15 |
| Discount | $\gamma$ | 0.99 |
| $\lambda$-target parameter | $\lambda$ | 0.95 |
| Actor gradient mixing | $\rho$ | 1 |
| Actor entropy loss scale | $\eta$ | $1 \cdot 10^{-3}$ |
| Actor learning rate | — | $4 \cdot 10^{-5}$ |
| Critic learning rate | — | $1 \cdot 10^{-2}$ |
| Slow critic update interval | — | 100 |
| Common | | |
| Policy steps per gradient step | — | 4 |
| MPL number of layers | — | 4 |
| MPL number of units | — | 400 |
| Gradient clipping | — | 100 |
| Adam epsilon | $\epsilon$ | $10^{-5}$ |
| Weight decay (decoupled) | — | $10^{-6}$ |

Table 9: We utilize the default training parameters specified in https://github.com/danijar/dreamerv2. Manipulating how the model is trained could help understand the sensitivity of the bound.

