# OpenReview forum: "Novelty Detection in Reinforcement Learning with World Models"
_ICML.cc/2025/Conference — ICML 2025 spotlightposter_

### Official Review · Reviewer_izpy · 2025-03-12

**Overall Recommendation:** 3

**Summary:**

This work proposes a novelty detection technique in model-based reinforcement learning called world models. It refers to sudden changes in the world model's visual properties or dynamic system as novelties. They use KL divergence between latent predictions with and without observable ground truth to design the novelty detection bound. The proposed technique is not dependent on additional hyper-parameters, and experiments are carried out on NovGrid, HackAtari, and RealWorldRL Suite.

**Claims And Evidence:**

Mostly, claims are supported by experimental results i.e.,

1. Even though Proposition 3.1 is supported by explanation and visualization of novelty detection bound in Figure 1. The quality of the paper can be improved further if authors come up with some guarantee on the bound.

2. Referencing Srivastava et al. 2014 for Proposition 3.2 is not enough, a little bit more explanation is needed to make the paper self-sufficient.

**Essential References Not Discussed:**

N/A

**Experimental Designs Or Analyses:**

The experimental designs for novelty detection of visual properties and dynamic systems are ok. Authors use the false positives rate, average delay error, AUC scores, and inference run-time speedup for evaluation of proposed novelty detection technique.

**Methods And Evaluation Criteria:**

The proposed novelty detection bound has been evaluated on NovGrid, HackAtari, and RealWorldRL Suite, these environments are suitable for the proposed method.

**Other Comments Or Suggestions:**

Typos or errors,
1. Ln 112 - col 1: signal.(Greenberg & Mannor, 2020) This
2. Ln 92 to 96 - col 1: complex sentence to get the meaning, so rephrase or at least give ref to eq 3.
3. Ln  215 - clo 2: (divergence of blue collapses to red)? or blue / differences collapse to orange?

**Other Strengths And Weaknesses:**

The main strength of the paper is that the proposed technique doesn’t require environment specific parameters/threshold tuning.

The main weakness, as noted above, is that the authors don’t provide rigorous proof of the novelty bound.

**Questions For Authors:**

Please see above, including the Claims and Evidence section.

**Relation To Broader Scientific Literature:**

Most of the prior novelty detection methods need environment specific parameter turning. The proposed method doesn’t need the environment specific parameters turning, which is a good advantage. Further, novelty detection is under explored in the model-based RL settings. Hence, this work is relevant to the community.

**Theoretical Claims:**

The claims look ok, apart from the issues listed in the Claims And Evidence section.

---

> ### Author Rebuttal · Authors · 2025-03-31
>
> We thank the reviewer for their suggestions on improving our work. In response, we outline provisional revisions below:
>
> First, we note that we have fixed each of the typos and syntax errors that were identified.
> Specifically, we have:
> * Moved the period to after the citation.
> * Rephrased Ln 92 to 96 - col 1 to:
> 	To develop the bound, we observe that, under nominal conditions, any divergence should be smaller than that of the predicted latent world state computed with the initial hidden state, as the latter prediction becomes increasingly inaccurate.
> * Rephrased Ln 215 - clo 2 to: (blue differences collapse to orange).
>
> Regarding the following statement:
> ```
> Referencing Srivastava et al. 2014 for Proposition 3.2 is not enough, a little bit more explanation is needed to make the paper self-sufficient.
> ```
> We first reword Proposition 3.2 to align with terminology used by other works [2]:
> *If the cross entropy score comparison becomes **nonnegative** when introducing the vector $x_t$, then 4 defines a decision boundary in the latent space over the measure of robustness to partial destruction of the input (Vincent
> et al., 2008; Srivastava et al., 2014), i.e.:*
>
> $KL[p_{\phi}(z_t|h_t,x_t)||p_{\phi}(z_t|h_t)] +
> KL[p_{\phi}(z_t|h_t,x_t)||p_{\phi}(z_t|h_0,x_t)]
> \leq\\
> KL[p_{\phi}(z_t|h_t,x_t)||p_{\phi}(z_t|h_0)]$.
>
> and have extended the derivations section in the appendix to include a more explicit tone:
>
> Let the full posterior $p_{\phi}(z_t|x_t,h_t)$ be the most informed distribution and the desired distribution with support over the latent space  $\mathrm{Z}$ at any given $t$, and let $f_\theta$ map to the representation of a distribution with the support of $\mathrm{Z}$ as well.
>
> Utilizing notation similar to that in Vincent et al. (2008), denote $(h_t,x_t)$ as the clean input $x$ (not to be confused with $x_t$), and define three variants of $x'$: $\left\lbrace x_{(-h_t)}', x_{(-x_t)}', x_{(-h_t,-x_t)}' \right\rbrace$, which denote the removal of features $(h_t)$, $(x_t)$, and $(h_t,x_t)$ from the clean input $x$, respectively.
>
> Since the loss is at the latent level, we measure the robustness to partial destruction criterion of a single input $x$ for a desired distribution as: $L(p_{\phi}(z_t|h_t,x_t), f_{\theta}(x'))$
>
> Therefore, if $L$ is chosen to be the KL divergence, and $x'$ becomes explicit, then we can derive Proposition 3.2 as:
>
> $L(p_{\phi}(z_t|h_t,x_t), p_{\phi}(z_t|h_0)) \geq L(p_{\phi}(z_t|h_t,x_t), p_{\phi}(z_t|h_t)) + L(p_{\phi}(z_t|h_t,x_t), p_{\phi}(z_t|h_0,x_t)) =$
> $$KL(p_{\phi}(z_t|h_t,x_t) \parallel p_{\phi}(z_t|h_0)) \geq KL(p_{\phi}(z_t|h_t,x_t) \parallel p_{\phi}(z_t|h_t)) + KL(p_{\phi}(z_t|h_t,x_t) \parallel p_{\phi}(z_t|h_0,x_t))$$
>
>
> where $p_{\phi}$ corresponds to the distribution modeled by $f_{\theta}$ given some $x^{'}$, and $h_0$ represents the state of $h$ when no conditioning information is available.
>
> ```
> The quality of the paper can be improved further if authors come up with some guarantee on the bound.
> ```
> Finally as suggested, we agree that we should attempt to give the reader intuition surrounding our method by extending the appendix with the following theoretical guarantee which shows that for a theoretical perfectly trained world model  (i.e: $z_t \perp x_t \mid h_t$ and there exists a set of parameters $\phi^* $ such that:
> $  p_\phi(\cdot|\cdot) = p^*(\cdot|\cdot)$
> where $p^*( \cdot | \cdot)$ denotes the well defined true conditional distribution) then EQ 4 holds if:
> $
> E_{p_\phi(z_t \mid h_t,x_t)}\left[\log \frac{p_\phi(z_t \mid h_0,x_t)}{p_\phi(z_t \mid h_0)}\right] \ge 0.
> $
> Which can be interpreted as holding if the Expected Information Gain (EIG) of $x_t$ is nonnegative under the distribution of $p_\phi(z_t \mid h_t,x_t)$.
>
> Ideally, although as an informal side note, we would like to intuitively interpret $KL[p(z_t |h_t x_t )|| p(z)]$ as proportional to $I(z_t;h_t,x_t)$ (which is often a substitution in practice [1]). Where
> * $KL[p(z_t |h_t, x_t )|| p(z_t)]$  is substituted by $I(z_t;h_t,x_t) = H(z_t) - H(z_t|h_t,x_t)$
> * $KL[p(z_t |h_t, x_t )|| p(z_t | h_t)]$ is substituted by $I(z_t; x_t | h_t) = H(z_t|h_t) - H(z_t|h_t,x_t)$
> * $KL[p(z_t |h_t, x_t) || p(z_t | x_t)]$ is substituted by $I(z_t;h_t | x_t) = H(z_t|x_t) - H(z_t|h_t,x_t)$
>
> Therefore, if showing (EQ 4) was somehow related to showing:
> $H(z_t) - H(z_t|h_t,x_t) \geq H(z_t|h_t)- H(z_t|h_t,x_t) + H(z_t|x_t)- H(z_t|h_t,x_t)$
>
> Then simplification would show:
> $H(z_t) - H(z_t|h_t) \geq H(z_t|x_t) - H(z_t|h_t,x_t) \implies$
>
> $I(z_t;h_t) \geq I(z_t;h_t|x_t) (*)$
> Where a potential key result would lie: EQ 4 holds if knowing X reduces the mutual information between Z and H.
> ```
> [1] Zhao, S., Song, J., & Ermon, S. (2019). InfoVAE: Balancing Learning and Inference in Variational Autoencoders. Proceedings of the AAAI Conference on Artificial Intelligence, 33(01), 5885-5892.
> [2] Vincent, P., Larochelle, H., Bengio, Y., and Manzagol, Extracting and composing robust features with denoising autoencoders. ICML 2008
> ```

---

### Official Review · Reviewer_oBkv · 2025-03-14

**Overall Recommendation:** 4

**Summary:**

The paper seeks to determine when there is novelty in an environment by using world models, and particularly when there is high prediction error with such world models. Such an approach aligns strongly with existing neuroscience work. The paper demonstrates with strong results their approach, exceeding the performance of most existing approaches.

## Update after rebuttal
The author's successfully responded to my questions and the questions of the other reviewers. I feel confident in my score of a 4 and in the broader merit of the paper.

**Claims And Evidence:**

The claims in the submission are very strongly supported by good evidence. The authors used strong baselines and benchmarks and I did not find any issues with any claims/experiments.

**Essential References Not Discussed:**

N/A

**Experimental Designs Or Analyses:**

I checked the validity of all experiments in the main paper and did not find any issues.

**Methods And Evaluation Criteria:**

The benchmark datasets and evaluation criteria used was standard and makes sense for the problem.

**Other Comments Or Suggestions:**

- The authors state "We introduce a novel technique for novelty detection without the need to manually specify thresholds, and without the need for additional augmented data." but then they later state "Our technique calculates a novelty threshold bound" so there seems to still be a reliance on the novelty threshold bound? I'm unsure clarification would be helpful.

**Other Strengths And Weaknesses:**

## Strengths
- The approach is very simple and elegant, aligning with neuroscience ideals of uncertainty/novelty detection guided by high prediction errors
- The results strongly support the claims of the authors, and the metric has several potential downstream use cases for world modeling, including determining whether the world model is properly using history (Figure 1) as well as whether the world model has converged or not.

## Weaknesses
- While the approach is simple, that also means the contribution is rather limited. I also see this as a small weakness
- The authors used an older version of dreamer (V2 instead of V3) which begs the question--does the proposed approach work for newer MBRL based approaches?
- The proposed approach is not compatible with raw inputs (i.e. when using diffusion) or non probabilistic latents. I also see this as a small weakness

**Questions For Authors:**

- Why is the ground truth for Table 1 so noisy?
- I don't understand this point "in nominal conditions, any divergence should always be smaller than the divergence of the predicted latent world state computed with the initial hidden state instead of the current initial hidden state; the latter prediction should be increasingly inaccurate." Can the authors elaborate on this?
- Why did they use DreamerV2 instead of the newer DreamerV3?

**Relation To Broader Scientific Literature:**

The paper is broadly related to world models, Reinforcement Learning and particularly Model Based reinforcement learning, as well as novelty detection/uncertainty.

**Theoretical Claims:**

I checked the correctness of all proofs in the main paper.

---

> ### Author Rebuttal · Authors · 2025-03-30
>
> Thank you for your review, hopefully we can interpret your questions and clarify as well as possible:
> * Why is the ground truth for Table 1 so noisy?
>    * The ground truth component of the table shows the true observation given to the agent, specifically the noisy observation is a sample from the noisy environment that occurs when we inject the novelty ‘noise high’ in the humanoid setting.
> * I don't understand this point "in nominal conditions, any divergence should always be smaller than the divergence of the predicted latent world state computed with the initial hidden state instead of the current initial hidden state; the latter prediction should be increasingly inaccurate." Can the authors elaborate on this?
>    * Thanks for pointing out that this statement might be confusing to the reader, with that statement we are attempting to describe our observation in Figure 1. In response we have rephrased the statement to be: “to develop the bound, we observe that, under nominal conditions, any divergence should be smaller than that of the predicted latent world state computed with the initial hidden state, as the latter prediction becomes increasingly inaccurate.”
> * The authors state "We introduce a novel technique for novelty detection without the need to manually specify thresholds, and without the need for additional augmented data." but then they later state "Our technique calculates a novelty threshold bound" so there seems to still be a reliance on the novelty threshold bound? I'm unsure clarification would be helpful.
>    * No problem. To clarify, we are stating that the user does not have to be mindful of explicitly/manually setting a measurement of uncertainty $\lambda$ beforehand to trigger the detection of novelty. Our technique will simultaneously handle the bound (as well as the detection), while minimizing false positives.
> * The authors used an older version of dreamer (V2 instead of V3) which begs the question--does the proposed approach work for newer MBRL based approaches? … Why did they use DreamerV2 instead of the newer DreamerV3?
>    * We consider our work on the V2 algorithm to be mainly based on the RSSM component of the world model, and although the major contributions of going from the Tensorflow V2 to the JAX V3 introduced faster optimizations, this move would have made the experimental design more complex despite both V2 and V3 being centered around the core RSSM framework. Therefore, for newer MBRL approaches, we believed it was more impactful to take time to test the newer Diamond (2024) [1], and IRIS (2023) [2] world models, which unlike DreamerV3 or V2, do not use the RSSM framework.
> * The proposed approach is not compatible with raw inputs (i.e. when using diffusion) or non probabilistic latents. I also see this as a small weakness.
>    * For diffusion based approaches, we found that the PP-MARE method (see Table 7, Section 4 PP-MARE, and Appendix B paragraph Diamond) could successfully be used directly on raw inputs. As for non-probabilistic latents, (possibly used if it is expected that the environment is deterministic and fully observable), then our setting of being in a stochastic partially observable setting no longer holds. However, we hypothesize that a subtle extension could be that the KL loss could be substituted with a binary operator (or suitable deterministic metric) over the posterior and prior representations, when conditioned on different information.
> * While the approach is simple, that also means the contribution is rather limited. I also see this as a small weakness
>    * Thanks for the feedback. We purposely crafted the main method to be simple, but to the best of our knowledge, we are the first to test novelty detection in both Atari and 3D-DMC environments entirely in the image space (see footnote 2). We also introduce PP-MARE for those who are interested in working outside of the latent space, and are okay with specifically defining their threshold.
> ```
> [1] Alonso, E., Jelley, A., Micheli, V., Kanervisto, A., Storkey, A., Pearce, T., and Fleuret, F. Diffusion for world modeling: Visual details matter in atari. 2024.
> [2] Micheli, V., Alonso, E., and Fleuret, F. Transformers are sample-efficient world models. In The Eleventh International Conference on Learning Representations, 2023.
> ```

---

> > ### Comment · Reviewer_oBkv · 2025-04-07
> >
> > [Accidentally posted as official comment]
> > The authors successfully responded to my comments and questions and I feel confident keeping my score as a 4, especially understanding better now that there are no bounds to be manually tuned which I believe is a strong advantage.

---

### Official Review · Reviewer_iyHf · 2025-03-23

**Overall Recommendation:** 4

**Summary:**

The work proposes a principled method for detecting novelty in RL agents that use latent dynamics models, such as DreamerV2. The central idea is that when an agent encounters novel observations or dynamics, the latent state inferred from the current observation (posterior) will significantly differ from the one predicted by its internal dynamics (prior). This mismatch is captured through the KL divergence between the posterior and prior latent distributions. Rather than relying on heuristic thresholds, the authors introduce theoretically motivated bounds that relate this KL divergence to priors conditioned on reduced information — namely, with no history or only the current observation. These bounds are formalized through Propositions 3.1 to 3.2, enabling novelty detection without supervision or tuning.

The paper evaluates this method across several domains, including DeepMind Control Suite, MiniGrid, and Atari, introducing both observation and transition-based novelties mid-trajectory. The results show that the proposed KL-based approach reliably detects novelty with low false positive rates and sharp response times, outperforming baselines like RIQN. A key contribution is the demonstration that a well-trained latent world model can serve as a robust internal monitor of novelty, offering a unified, efficient, and interpretable mechanism to detect both perceptual and dynamic anomalies in reinforcement learning environments.

### Update after rebuttal

I am confident in my assessment of accepting this paper, It's a good work.

**Claims And Evidence:**

The claims made in the submission are largely supported by clear and convincing empirical evidence. The authors claim that their KL-based novelty detection method, grounded in the latent space of a Dreamer-style world model, is capable of reliably identifying both observation-level and dynamics-level novelties across a range of environments. This is substantiated through a series of well-designed experiments on DMC, MiniGrid, and Atari, where the proposed method outperforms baselines. The per-timestep KL score clearly spikes in response to injected novelties (can be seen even during training). Tables 2-6 show clear quantitative results where the method is able to almost instantaneously detect a novelty in most cases.

That said, while the empirical results are strong, some of the theoretical claims—particularly Propositions 3.2 —rest on assumptions about the behavior of KL divergence that may not always hold in practice (discussed later), such as its non-metric nature and dependence on variational approximations. This might not be a major flaw but should be talked about in the paper in my opinion. Nonetheless, the main claims regarding the method’s novelty detection performance are well-supported by the presented evidence across a variety of setups.

**Essential References Not Discussed:**

None to the best of my knowledge.

**Experimental Designs Or Analyses:**

The experimental design and analysis appear generally sound and well-aligned with the paper’s goals. The authors evaluate their method across three diverse benchmarks—MiniGrid, DMC, and Atari—with injected observation and transition novelties that simulate realistic distribution shifts. The novelty is introduced mid-trajectory, allowing for precise analysis of detection timing. The use of per-timestep KL divergence as a novelty score, along with aggregated trajectory-level evaluations (e.g., AUC, false alarm rate, detection delay), provides a decent measure of novelty detection. The comparison against multiple baselines is thorough ensuring a fair and comprehensive evaluation.  Overall, the key metrics and qualitative analyses convincingly support the claims, and no major flaws are apparent in the experimental methodology.

**Methods And Evaluation Criteria:**

Yes, the paper uses a sufficient set of baselines, datasets, and methods for the problem at hand. The baselines are good, covering a broad spectrum of novelty detection strategies. The evaluation spans three well-established benchmarks—MiniGrid, DeepMind Control Suite, and Atari—which together capture a wide range of perceptual and dynamic challenges. The proposed method is integrated naturally into DreamerV2, and its performance is compared rigorously across these environments, demonstrating the method’s generality and robustness. They also test the effectiveness of their approach on other world model approaches and the algorithm works well (PP-Mare variant) on diffusion based but struggles on Transformer based which was also an observation by a previous work.

An important issue here is: **authors mention an Appendix H but there is none present**, that possibly discussed transformer based world models and KL bound for novelty detection. I will recommend authors to add that for more context.

**Other Comments Or Suggestions:**

I think Appendix H is missing (or that is meant to be linked with appendix B, IRIS).

I think the discussion on exploration should be included in the main paper and "Detection Score Trend Over Time" + Figure 2 can be shifted to appendix. An under-explored policy directly impacts the reliability method. Additionally, as future work, authors may want to incorporate uncertainty-aware priors to distinguish novelty from ignorance to tackle insufficient exploration (though I understand, that possibly means introducing changes to the world model, which is not the point of the paper.)

**Other Strengths And Weaknesses:**

I don't have many concerns barring a suggestion to include some theoretical clarifications which I stated earlier as well, I think the paper is well written and evaluations are exhaustive. The need to not tune per environment makes the method extremely versatile. The reproducibility is as good as it can be.

However, one important weakness I think is: the approach's results will be as good as the world model themselves, i.e., their applicability is limited in cases where interactions with the environment is expensive or potentially dangerous.

**Questions For Authors:**

None

**Relation To Broader Scientific Literature:**

The paper builds directly on the growing body of work in model-based reinforcement learning, particularly approaches using latent world models such as DreamerV2, and extends these frameworks to address the underexplored problem of unsupervised novelty detection. While prior works like Dreamer and PlaNet focus on learning compact latent dynamics for control and planning, they do not consider the challenge of identifying distributional shifts during deployment. This paper shows that the KL divergence between the posterior and prior latent distributions—already computed internally by such models—can be leveraged as an effective and lightweight novelty signal. Importantly, unlike reconstruction- or prediction-based baselines, this method operates entirely in the latent space, making it more stable and closely aligned with the agent's internal belief state.

Relative to traditional novelty detection methods such as autoencoders, latent prediction models, or uncertainty-based approaches like RIQN, the proposed method introduces a novel theoretical framework based on KL divergence bounds derived from priors conditioned on partial information (e.g., without memory or observation). These bounds are used to derive threshold-free detection criteria rooted in variational inference, inspired by dropout-based generalization tests. A key advantage of this approach is that it requires **no additional tuning or threshold hyperparameters**, making it both more interpretable and easier to deploy in practice.

**Theoretical Claims:**

The paper includes several theoretical claims in the form of Propositions 3.1 and 3.2, which are used to justify bounds for novelty detection based on KL divergence between posterior and prior latent state distributions. The propositions are intuitive and well-motivated, but some rely on assumptions that are not fully formalized.

While Proposition 3.2 presents an intuitive inequality that resembles a triangle inequality over KL divergences, it is important to note that **KL divergence is not a true metric and does not satisfy the triangle inequality** in general. The bound in Proposition 3.2 compares the posterior to three different priors—one conditioned on memory, one on the current observation, and one on neither—and assumes that if the posterior is close to each of the simplified priors individually, then it should also be close to the full prior. However, this assumption is structurally similar to triangle inequality reasoning, which is not formally valid in KL space due to its asymmetry and lack of metric properties. As a result, the inequality may not hold even when all distributions involved are close in distributional terms. This weakens the theoretical guarantee of the proposition and suggests that violations of the bound might occur not only due to novelty, but also due to the non-metric behavior of KL divergence, particularly in high-dimensional or undertrained latent spaces. However, empirically, I don't think this is a big issue but I will still recommend authors to discuss a bit more on this, especially in the derivations.

---

> ### Author Rebuttal · Authors · 2025-04-01
>
> We thank the reviewer for their time in working to improve our presentation and their encouraging feedback. We are delighted that you took the time to assist the quality of our work. In response, we outline provisional revisions below:
> * We agree that the discussion of C.2 may be more informative to the reader than the section *Detection Score Trend Over Time*, since the world model's final performance is directly tied to the exploration of the environment. Therefore, we have moved section C.2, Figure 7, and Table 8 into the main manuscript, and have moved *Detection Score Trend Over Time*, and Figure 2 into the appendix for those interested in visualizing the anomaly score trend. Section C.2 now appears after the section *alternative world models*.
> * We agree that it is important to make a note to the reader about the downsides of not using a proper metric such as the KL divergence, and the method's dependence on variation approximations. In response, we've extended the derivations section to discuss explicit details of the dependence on variational approximations as well as assumptions on the properties of the model itself. We state: “We briefly note that KL divergence is not a true metric. The bound in Proposition 3.2 compares the posterior to three different priors and provides a useful approximation for detection, rather than a strict mathematical guarantee.
> Formally, for distributions P, Q, and R, the KL divergence does not generally satisfy $D_{KL}(P \parallel R) \leq D_{KL}(P \parallel Q) + D_{KL}(Q \parallel R)$. This means Eq 4 might be violated not only due to novelty but potentially due to the inherent non-metric behavior of KL divergence, particularly in high-dimensional or complex/untrained latent spaces (see exploration effects)”. In addition, we also redirect the reader to this discussion as we note the reliance of functional approximators in the limitations section, which reads:
> *While our proposition using KL divergence has proven effective empirically, future work might explore alternative divergence measures with metric properties, such as the Jensen-Shannon divergence or Wasserstein distance, which could provide stronger theoretical guarantees. Additionally, exploring the relationship between latent space dimensionality and the reliability of variational approximations (of which our bound is based on see Appendix D) could yield insights into optimizing model architecture for novelty detection.*
> * We have summarized the contents of Appendix H in Appendix B to improve the self sufficiency of the paper, (which ties into C.2 Exploration effects) of which we discuss the exploration trick of weighing a single action heavily that IRIS incorporated to successfully earn a higher score (which could affect the performance of the world model). Specifically, we add to Appendix B:
> *Although our initial interpretation had moderate success, we expect that future work should be able to improve by adjusting the bound as well as addressing the exploration concerns given the exploration trick used in IRIS, of which the "up" action was stated to be heavily oversampled in favor of improving the agent performance, rather than improving the world model performance. Ultimately, we leave the task of enhancing novelty detection in the IRIS world model for future work.*

---

> > ### Comment · Reviewer_iyHf · 2025-04-02
> >
> > Thank you for incorporating these changes. I am excited to read your new manuscript. I keep my score.

---

### Decision · Program_Chairs · 2025-05-01

**Decision:**

Accept (spotlight poster)

**Comment:**

This paper proposes a theoretically motivated approach for novelty detection in model-based reinforcement learning by utilizing the KL divergence between latent posterior and prior distributions within world models. Across all three reviewers, there is agreement that the method is simple, elegant, and strongly supported by empirical evidence across diverse benchmarks. Theoretical propositions are appreciated for their intuition, though some reviewers noted that assumptions about KL divergence’s properties may weaken the formal guarantees--an issue the authors addressed thoughtfully in the rebuttal. The approach’s key strengths include threshold free deployment, strong detection performance, and a clear link to internal model uncertainty. While some concerns were raised about limited applicability to newer architectures (like DreamerV3 or non-probabilistic latents), these were adequately acknowledged and contextualized in the response. Overall, the paper received solid endorsements, with all reviewers recommending acceptance. Please consider adding impact statement in final version.